# Lignocellulolytic Enzyme Production from Wood Rot Fungi Collected in Chiapas, Mexico, and Their Growth on Lignocellulosic Material

**DOI:** 10.3390/jof7060450

**Published:** 2021-06-05

**Authors:** Lina Dafne Sánchez-Corzo, Peggy Elizabeth Álvarez-Gutiérrez, Rocío Meza-Gordillo, Juan José Villalobos-Maldonado, Sofía Enciso-Pinto, Samuel Enciso-Sáenz

**Affiliations:** 1National Technological of Mexico-Technological Institute of Tuxtla Gutiérrez, Carretera Panamericana, km. 1080, Boulevares, C.P., Tuxtla Gutiérrez 29050, Mexico; D18270772@tuxtla.tecnm.mx (L.D.S.-C.); peggy.ag@tuxtla.tecnm.mx (P.E.Á.-G.); rocio.mg@tuxtla.tecnm.mx (R.M.-G.); juan.vm@tuxtla.tecnm.mx (J.J.V.-M.); 2Institute of Biomedical Research, National Autonomous University of Mexico, Circuito, Mario de La Cueva s/n, C.U., Coyoacán, México City 04510, Mexico; sofienciso@hotmail.com

**Keywords:** fungi, ligninolytic enzymes, cellulase, xylanase, lignocellulosic biomass, manganese peroxidase, lignin peroxidase

## Abstract

Wood-decay fungi are characterized by ligninolytic and hydrolytic enzymes that act through non-specific oxidation and hydrolytic reactions. The objective of this work was to evaluate the production of lignocellulolytic enzymes from collected fungi and to analyze their growth on lignocellulosic material. The study considered 18 species isolated from collections made in the state of Chiapas, Mexico, identified by taxonomic and molecular techniques, finding 11 different families. The growth rates of each isolate were obtained in culture media with African palm husk (PH), coffee husk (CH), pine sawdust (PS), and glucose as control, measuring daily growth with images analyzed in ImageJ software, finding the highest growth rate in the CH medium. The potency index (PI) of cellulase, xylanase, and manganese peroxidase (MnP) activities was determined, as well as the quantification of lignin peroxidase (LiP), with the strains *Phlebiopsis flavidoalba* TecNM-ITTG L20-19 and *P**hanerochaete sordida* TecNM-ITTG L32-1-19 being the ones with the highest PI of hydrolase activities with 2.01 and 1.83 cellulase PI and 1.95 and 2.24 xylanase PI, respectively, while *Phlebiopsis flavidoalba* TecNM-ITTG L20-19 and *Trametes sanguinea* TecNM-ITTG L14-19 with 7115 U/L LiP activity had the highest oxidase activities, indicating their ability to oxidize complex molecules such as lignin.

## 1. Introduction

Lignocellulolytic enzymes capable of degrading hardwoods play an important ecological role as degraders of lignocellulosic biomass, in addition to having multiple applications in the textile, pharmaceutical, paper, food, cosmetic, environmental, biotechnological, detergent, and biofuel industries [1,2].

The importance of lignocellulolytic enzymes and the diversity of fungal species that produce them focus interest on the evaluation of their applications using agroindustrial waste that is disposed of without treatment as inducers for enzyme production.

Currently, the development of new technologies has been proposed to remove these pollutants from the environment. For this reason, numerous biological techniques involving organisms such as fungi have been developed for the removal of organic pollutants.

Among the fungi of interest are lignocellulolytic fungi, which are those that use lignocellulosic material as a source of carbon and energy. It is important to note that lignocellulosic materials are mainly composed of cellulose, hemicellulose, and lignin [1]; the latter being of great interest since it is a complex compound that has multiple aromatic rings in its structure. Fungi have developed a collection of extracellular oxidative enzymes to degrade lignin. They employ different types of peroxidases, including lignin peroxidase (LiP), manganese peroxidase (MnP), and versatile peroxidases (VP). While some of these enzymes are capable of degrading lignin or lignin fragments, peroxidases also degrade lignin through oxidative mediators, small oxidizing agents that can penetrate the branched lignin polymer to trigger depolymerization. Lignocellulolytic fungi degrade cellulose, hemicellulose, lignin (main wood compounds), lignocellulosic residues, and a variety of environmental pollutants to use them as a source of carbon and energy, due to their secretion of extracellular hydrolase and oxidase enzymes such as cellulase, xylanase, manganese peroxidase, laccase and lignin peroxidase [3,4].

The fungi that cause white wood rot are characterized by possessing a group of enzymes capable of degrading wood lignin known as ligninases or ligninolytic enzymes. Lignin is what gives wood its characteristic brown color, which when degraded by these fungi is enriched in cellulose whose color is white, hence the name of this type of rot. There are two groups of white-rot fungi: the simultaneous white-rot fungi consume lignin and carbohydrates rather parallelly and to a similar extent, whereas the selective white-rot species at least in the beginning prefer the lignin [5]. Within this group of ligninolytic enzymes, there are hydrolases, oxidases, and peroxidases whose function is to begin the consumption of lignin through strong oxidations. It is important to mention that the reactions carried out by these enzymes are nonspecific since they oxidize lignin and a variety of aromatic compounds [6,7,8].

Therefore, this work aims to evaluate the production of lignocellulolytic enzymes from eighteen isolates of lignocellulolytic fungi collected in Chiapas, Mexico, and to analyze their growth on lignocellulosic material, providing perspectives for research regarding the degradation of environmental pollutants and agroindustrial residues as well as their biotechnological applications in obtaining by-products with added value.

## 2. Materials and Methods

### 2.1. Organisms

Specimens of white and brown wood rot fungi were collected from three different sites in the state of Chiapas, Mexico: (1) San José Educational Park (SJEP) (16°43′26.54″ N and 92°41′55.55″ W), located at an altitude between 2350 and 2380 m above sea level (masl), with a predominantly humid temperate climate with summer rains, mean annual temperature of 25.1 °C, annual precipitation of approximately 1276 mm, and vegetation consisting of pine–oak forests; (2) Laguna Verde Ecotourism Park (LVEP) (17°07′56.37″ N and 93°09′48.77″ W), with an altitude between 1599 and 1631 m above sea level, semi-warm sub-humid climate with summer rains, annual rainfall of 1800 mm, average annual temperature of 27.3 °C and medium rainforest vegetation; (3) Facilities of the National Technological of Mexico-Technological Institute of Tuxtla Gutiérrez (TecNM-ITTG by its Spanish acronym) (16°45′27.63″ N and 93°10′21.75″ W), at 577 masl, warm sub-humid climate with summer rains, mean annual temperature of 33.2 °C, average annual rainfall of 900 mm and predominant vegetation of low deciduous forest [9,10,11,12]. For isolation and propagation, the collected macromycetes were disinfected, washing them with water to remove the residues of the substrate, and subsequently three washes were performed: sodium hypochlorite (5%) for 1 min, distilled water (to remove chlorine residues) for 1 min and ethanol (70%) for 1 min, at once, in the laminar flow bell, the macromycetes were rinsed with sterile distilled water. With a sterile scalpel, a vertical cut was made in the carpophore, and with a sterile dissection clamp the tissue was recovered and inoculated in solid medium of potato-dextrose agar (PDA) to obtain mycelium, at a temperature of 28 ± 1 °C. Each sample was assigned a key for identification.

### 2.2. Identification

Specimens were taxonomically identified by macroscopic and microscopic characteristics and dichotomous keys [13,14,15,16,17].

To carry out molecular identification, total genomic DNA was extracted from fungal isolates with a miniprep kit (ZYMO RESEARCH, California, USA), according to manufacturer’s instructions. Genomic DNA was amplified using the fungal specific primers ITS 4 (5-TCCTCCGCTTCTTATTGATATATGC-3) and ITS 5 (5-CTTGGTCATTTAGAGGAAGTAA-3) [18]. Amplification was performed on a thermal cycler (SensoQuest, Göttingen, Germany) programmed as follows: 92 °C for 5 min, 35 cycles at 92 °C for 30 s, 52 °C for 30 s, 72 °C for 45 s, followed by a 5 min extension step at 72 °C. PCR products were electrophoresed at 85 V on 1% agarose gels and the resulting bands were observed on a UV transilluminator (BIO-RAD, Hercules, CA, USA). The amplifications were sequenced at Macrogen Laboratories (Korea). Once the sequences were obtained, they were analyzed and nucleotide sequence similarity search was performed in GenBank using the Basic Local Alignment Search Tool (BLAST) [19], from the National Center for Biotechnology Information (NCBI).

### 2.3. Isolates Growth in Solid Culture Media

For the evaluation of fungal mycelial growth, a modified mineral salt solution was added to all the culture media [20], containing (g/L): NaNO_3_ (18), KH_2_PO_4_ (1.3), MgSO_4_·7H_2_O (0.38), CaSO_4_·2H_2_O (0.25), CaCl_2_ (0.055), FeSO_4_·7H_2_O (0.015), MnSO_4_·H_2_O (0.012), ZnSO_4_·7H_2_O (0.013), CuSO_4_·7H_2_O (0.0023), CoCl_2_·6H_2_O (0.0015) and H_3_BO_3_ (0.0015) and phosphate solution (K_2_HPO_4_ 45.6 g/L and KH_2_PO_4_ 27.2 g/L), in addition to a carbon source.

Three agroindustrial substrates were selected as a carbon source: African palm husk (PH) from *Elaeis guineensis* (belonging to the residue in the oil process of the company OLEOSUR S. A. de C.V. located in the municipality of Acapetahua, Chiapas, Mexico), coffee husk (CH) from *Coffea arabica* (from waste from the company PRANA, located in the municipality of Tuxtla Gutiérrez, Chiapas, Mexico) and pine sawdust (PS) from *Pinus* spp. The substrates were ground and sieved in a No. 100 mesh, particle size 0.149 mm. For the medium, 2% agar and 5% agroindustrial substrate were added as carbon source and the isolated fungi were inoculated and incubated at a temperature of 28 ± 1 °C [21].

The control medium was prepared with 2% agar and 10 g/L of glucose (control) as a carbon source, and this medium was inoculated with the isolated fungi and incubated at a temperature of 28 ± 1 °C.

The mycelial growth rate (mm/d) was measured every 24 h until the end of the exponential growth phase of each strain using images analyzed with ImageJ software (Image Processing and Analysis in Java, National Institutes of Health, Bethesda, MD, USA), using a vernier as a scale [22].

### 2.4. Cellulase Activity

For semi-quantitative test of cellulase activity, solid media were prepared with 2 g/L carboxymethyl cellulose, 2% bacteriological agar, and a salt solution containing (g/L): NaNO_3_ (2), K_2_HPO_4_ (1), MgSO_4_-7H_2_O (0.5), and KCl (0.5). The medium was inoculated with the isolated strains and incubated at 28 ± 1 °C. After the incubation time, in the exponential growth phase of each fungus, 10 mL of 0.8% Lugol’s iodine was added as a developer until the agar surface was covered, allowed to stand for 5 min and the Lugol’s iodine was redrained, then the mycelial growth and the halo of enzymatic activity by the change in coloration of the medium were measured by images analyzed in ImageJ software using a vernier as a scale [23,24,25,26].

### 2.5. Xylanase Activity

For semi-quantitative test of xylanase activity, solid media were prepared with 2 g/L birch xylan, 2% bacteriological agar, and a salt solution containing (g/L): NaNO_3_ (2), K_2_HPO_4_ (1), MgSO_4_-7H_2_O (0.5), and KCl (0.5). The medium was inoculated with the isolated fungi and incubated at 28 ± 1 °C. After the incubation time, in the exponential growth phase of each fungus, 10 mL of 0.8% Lugol’s iodine was added as a developer until the agar surface was covered, allowed to stand for 5 min and the Lugol’s iodine was removed, then xylanase activity was determined by measuring the diameter of the mycelial growth and hydrolysis halo of enzyme activity by images analyzed in ImageJ software using a vernier as a scale [23,26].

### 2.6. Manganese Peroxidase Activity

For the semi-quantitative manganese peroxidase activity, solid media were prepared with PDA 39 g/L and phenol red 0.5 g/L as an oxidative indicator, since the oxidation of the latter by the action of the enzyme produces a discoloration reaction from red to orange. The medium was inoculated with the isolated fungi and incubated at 28 ± 1 °C. After the incubation time, in the exponential growth phase of each fungus, the mycelial growth and enzyme activity halo (color change of the medium due to the oxidation of phenol red) were measured employing images analyzed in ImageJ software using a vernier as a scale [21,26,27].

The following formula was used to measure the enzyme potency index of cellulase, xylanase, and manganese peroxidase activities:(1)$$\mathrm{PI}=\frac{\mathrm{HA}}{\mathrm{HM}}$$

PI = enzyme potency index; HA = halo diameter of the enzyme activity; HM = halo diameter of fungal mycelium.

### 2.7. Lignin Peroxidase Activity

The medium in which the fungi had the highest growth from which the enzyme extract for the liquid reactions was obtained was inoculated with the isolates and incubated at 28 ± 1 °C. After the incubation time, during the exponential growth phase of each fungus, the lignin peroxidase enzymatic activity was determined, using ABTS as an oxidizing agent that produces a color change from colorless to blue-green. Liquid reactions with a total volume of 300 µL were prepared in triplicate. Each reaction mixture contained 171 µL of 100 mM acetate buffer pH3, 43 µL of 10mM ABTS as an oxidizing agent, 43 µL of enzyme extract previously filtered and centrifuged at 8000× *g* for 5 min, and 43 µL of hydrogen peroxide. The blank reaction contained 214 µL of 100 mM acetate buffer pH3, 43 µL of enzyme extract, and 43 µL of hydrogen peroxide. One blank was made for each sample. Once the reaction mixture was prepared, it was incubated at 30 °C for 1 h in a Lumistell IEC-42c incubator, after which time the reaction was stopped by placing the samples in the cold, and absorbance readings were immediately taken on a NanoDrop ONE at 420 nm [28,29].

The lignin peroxidase enzyme activity was calculated with the following formula:(2)$$\mathrm{EA}\left(\frac{U}{L}\right)=\frac{a*\mathrm{TV}}{\varepsilon *l*\mathrm{EV}*T}$$
EA = enzymatic activity expressed in units per liter; U = moles ABTS oxidized per minute; a = absorbance at given λ; TV= total reaction volume; ε = molar extinction coefficient at that λ (36M^−1^cm^−1^); l = optical path (1 cm); EV = enzyme extract volume; T = time.

One unit of enzyme activity is defined as the amount of enzyme capable of oxidizing 1 mol of ABTS per minute.

All assays were performed in triplicate.

### 2.8. Experimental Design

To compare the effect of the different culture conditions, a completely random design was used by obtaining four different treatments, where the carbon source was varied, and the media used were: PH, CH, PS and glucose, three repetitions were performed for each treatment.

### 2.9. Statistical Analysis

For statistical analysis, we performed the analysis of the variance (ANOVA) and the test analysis of Tukey (*p* < 0.05). The Statgraphics XVI.II centurion program was used.

## 3. Results

### 3.1. Organisms

From the three sites SJEP, LVEP, and TecNM-ITTG, 33 fungi specimens were collected. Among the specimens collected were brown-rot fungi and white-rot fungi, among other fungi. Of the 33 specimens collected, 20 strains were isolated and propagated in PDA solid culture medium at 28 ± 1 °C. The collected specimens were deposited in the CHIP Herbarium of the Jardín Botánico Dr. Faustino Miranda, SEMAHN.

The origin trees from which the isolates from SJEP were derived were *Pinus ayacahuite*, *Pinus strobus*, *Pinus teocote*, *Pinus montezumae*, *Pinus oocarpa*, *Quercus oleoides* and *Quercus chartacea*; from LVEP were *Pinus* spp. and *Quercus* spp.; and the trees from which the isolates from Facilities of TecNM-ITTG were derived were *Mangifera indica*.

### 3.2. Identification

After carrying out the taxonomic and molecular identification, eighteen species were identified that correspond to the taxonomic category shown in Table 1, indicating the site where they were collected and the rot type.

### 3.3. Solid Media Growth

All species grew on the lignocellulosic media with the agroindustrial substrates as a carbon source; however, the highest growth rate was found on the coffee husk medium (Figure 1).

According to the statistical analysis, strain *Trichoderma harzianum* TecNM-ITTG L23-19 presents a statistically significant difference concerning the others, since it had the highest growth rate in the medium with coffee husk (30 mm/d), while strain *Curvularia spicifera* TecNM-ITTG L2-2-19 (2 mm/d) showed the lowest growth rate, followed by *Neopestalotiopsis macadamiae* TecNM-ITTG L27-2-19 (2.25 mm/d).

### 3.4. Cellulase Activity

Eighteen isolates were evaluated, and the statistical analysis indicates that strain *Phlebiopsis flavidoalba* TecNM-ITTG L20-19 presents a statistically significant difference concerning the others, since it had the highest cellulase activity potency index (2.01), while strain *Hyphodermella rosae* TecNM-ITTG L2-1-19 (0.436) was the one with the lowest cellulase activity PI; on the other hand, six of the strains evaluated (*Trichoderma citrinoviride* TecNM-ITTG L33-2-19, *Trichoderma harzianum* TecNM-ITTG L23-19, *Trichoderma longibrachiatum* TecNM-ITTG L30-2-19, *Trichoderma reesei* TecNM-ITTG L4-2-19, *Neopestalotiopsis macadamiae* TecNM-ITTG L27-2-19 and *Phanerochaete australis* TecNM-ITTG L4-1-19) did not present cellulase activity (Figure 2).

### 3.5. Xylanase Enzyme Activity

Statistical analysis indicates that of the eighteen isolates evaluated, *Phanerochaete sordida* TecNM-ITTG L32-1-19 presents a statistically significant difference concerning the others, being the strain with the highest xylanase activity potency index (2.239), while the strain with the lowest potency index was *Coprinellus disseminatus* TecNM-ITTG L9-1-19 (0.470) (Figure 3). Six of the strains evaluated did not show xylanase activity, the same strains that did not have cellulase activity.

### 3.6. Manganese Peroxidase Activity

Eighteen isolates were evaluated, and statistical analysis indicates that strain *Phlebiopsis flavidoalba* TecNM-ITTG L20-19 presents a statistically significant difference to the others, being the strain with the highest potency index of manganese peroxidase activity (1.673), which is a white-rot fungus, while the strain with the lowest potency index was *Trichoderma citrinoviride* TecNM-ITTG L33-2-19 (0.664) (Figure 4). Fifteen percent of the strains evaluated showed no manganese peroxidase activity (*Trichoderma reesei* TecNM-ITTG L4-2-19, *Phanerochaete australis* TecNM-ITTG L4-1-19 and *Epicoccum sorghinum* TecNM-ITTG L15B-19).

### 3.7. Lignin Peroxidase Activity

The lignin peroxidase enzyme was quantified in the eighteen isolates evaluated from liquid media with coffee husk, in which the fungi had the highest growth. According to the statistical analysis, the strain *Trametes sanguinea* TecNM-ITTG L14-19 presents a statistically significant difference concerning the others, which had the highest amount of lignin peroxidase enzyme (7115.226 U/L), which is a white-rot fungus, while the strain with the lowest amount of enzyme was *Trichoderma longibrachiatum* TecNM-ITTG L30-2-19 (540.123) U/L (Figure 5).

## 4. Discussion

Of the total number of strains identified, 89% are new records for the state of Chiapas, according to information from the catalog of the National Commission for the Knowledge and Use of Biodiversity (CONABIO by its Spanish acronym) and various publications and research carried out. The total number of new records indicates the importance of complete mycological studies for the state of Chiapas.

Wood-rot fungi are primarily responsible for the decomposition of lignocellulose, the most recalcitrant molecules in wood, and thus are critical in nutrient and carbon cycling in forest ecosystems. Wood-decomposing agaricomycetes produce three basic types of decomposition based on complex enzyme systems: white, brown, and soft rot [30]. Of the species identified, seven belong to the class of agaricomycetes, of which six are from the order of polypores and one from the order of Agaricales.

The common characteristic of white-rot fungi is the extensive degradation of lignin resulting in a whitish appearance of rotted wood. White-rot fungi comprise numerous fungi, mostly Basidiomycota belonging to the orders Polyporales and Agaricales. During wood decay, degradation occurs at a great distance from the hyphae by the diffusion of enzymes; the mechanism of action of the enzymes involved and the patterns of wood decay have been extensively studied since the late 1970s [31].

One characteristic of some Agaricomycetes is wood decay, especially white rot. Although several microorganisms degrade the polysaccharides (cellulose and hemicellulose) contained in woody biomass, the brown-rot and white-rot Agaricomycetes are considered key players in wood decay in nature. Brown-rot fungi depolymerize crystalline cellulose by cellulolytic enzymes and non-enzymatic mechanisms, and white-rot fungi are the species that almost exclusively biodegrade wood lignin in nature.

Of the wood decay fungi identified, one belongs to the order Agaricales, basidiomycetes fungi reported for the treatment of aromatic compounds such as nonylphenol, where the fungal degradation capacity is related to the secretion of ligninolytic enzymes such as laccase and MnP [32]. In turn, six species of the order Polyporales were identified (*Hyphodermella rosae* TecNM-ITTG L2-1-19, *Phanerochaete australis* TecNM-ITTG L4-1-19, *Phanerochaete sordida* TecNM-ITTG L32-1-19, *Phlebiopsis flavidoalba* TecNM-ITTG L20-19, *Trametes cingulata* TecNM-ITTG L13-19 and *Trametes sanguinea* TecNM-ITTG L14-19), species of fungi belonging to the order Polyporales, and they excrete high levels of the enzyme laccase and MnP [33]. The Polyporaceae family is one of the families that has numerous wood-degrading fungal species that have been thoroughly investigated for the production of lignin-modifying enzymes and their potential for industrial applications.

Thirty-nine classes of fungi found in waste degradation were reported, among which the Sordariomycetes class stands out, as species of this class were identified as the main cellulase producers. In addition to possessing genes that degrade lignocellulose, Sordariomycetes are efficient lignocellulose decomposers that can produce considerable amounts of lignocellulosic enzymes [34]. Among the fungi studied, six were found to belong to the class of Sordariomycetes within the order Hypocreales and Xylariales with four and two species of each, respectively. However, it is important to note that the order Xylariales have been reported as pathogenic fungi [35], belonging to the class Sordariomycetes, while other endophytic fungi also from the order Xylariales have been reported with antagonistic activity against phytopathogenic fungi [36].

Five fungi of the class Dothideomycetes, of the order Botryosphaeriales, Pleosporales, and Valsariales, were identified. Dothideomycetes are one of the most important classes of ascomycete fungi and comprise an incredible diversity of natural habitats [37].

Species of the genera *Coprinus*, *Panaeolus*, *Schizophyllum*, *Psilocybe*, *Trametes*, *Polyporus* and *Pycnoporus* have been reported as indicators of anthropogenic disturbance, and together with *Ganoderma* and *Rigidoporus* are typical species of tropical or humid subtropical forests, which is an ecological indication that the LVEP zone is a transitional area or an area with a certain degree of disturbance, since they tend to displace tropical forest species and to fruit in places used by humans as cultivated areas. While the families Agaricaceae, Entolomataceae, Tricholomataceae and Lepiotaceae can be considered as indicators of disturbance because they require certain very particular conditions for their development, it can be pointed out that some species of these families are located in areas disturbed by agriculture and/or cattle ranching, which were identified in this site, where a mixture of different forms of land use was observed, affecting mycological diversity, as in the case of the Coapilla forest. The Agaricaceae and Polyporaceae families are the most abundant in tropical zones and were identified within the TecNM-ITTG facilities.

The tropical species that were identified from the collection made in the LVEP, that have also been found in pine–oak forests, perhaps it is due it is a transition zone between different types of vegetation, or the thermophilic influence of the Pacific Ocean slope originated by its geographic position. Besides, fungal distribution is closely related to the ability to develop under different abiotic conditions (temperature, humidity, pH), mechanisms of dormancy, and spore dispersal, and taking into account that the macromycetes’ mycelia can range from millimeters to entire landscapes, it is difficult to determine the area occupied by each individual, which makes it impossible to measure the different parameters of biodiversity [38].

Fungi of the genus *Trametes* have been reported as medicinal mushrooms [38]. Identified species of these families were collected from the SJEP and TecNM-ITTG.

The most abundant fungi growing on organic matter reported for SJEP belong to the Polyporaceae family. Meanwhile, saprophytic and mycorrhizal fungi are important organisms for the ecological balance of the forest because they decompose organic matter and degrade cellulose, hemicellulose, and lignin in the ecosystems, and contribute to the formation of humus and the soil remineralization process. Agaricales and Polyporales fungi can adapt to changing conditions of temperature, humidity, and rainfall through various dispersal strategies adapted to rain and wind, indicating that they are more suitable for environments with moderate disturbance or disturbance [10], and of those collected at this site, three strains identified belong to the order of Agaricales and Polyporales.

The 18 isolates evaluated grew in the media with lignocellulosic material containing lignin among other compounds, and pine sawdust represents a reference since the fungi were collected from pine tree trunks; however, the medium with coffee husk propitiated that the growth rate in 16 of the 18 isolates evaluated was higher than in the medium with glucose (control), which is an easily accessible carbon source. This shows that the isolates are capable of taking as a carbon source the compounds contained in the medium, such as cellulose, pentoses, hexoses, melanoidin, and complex compounds that contain aromatic rings in their structure, such as chlorogenic acid, caffeine, lignin, caffeic acid, phenolic compounds, etc., to transform them and use them as a source of energy.

The strain *Trichoderma harzianum* TecNM-ITTG L23-19 presented the highest growth rate in the medium with coffee husk and expressed the enzymes manganese peroxidase and lignin peroxidase, which are involved in degradation reactions of complex compounds, and by not expressing the enzymes xylanase and cellulase hydrolases, this indicates evidence of a probable more specific metabolism to degrade more complex molecules such as aromatic compounds present in the medium.

The strains *Curvularia spicifera* TecNM-ITTG L2-2-19 and *Coprinellus disseminatus* TecNM-ITTG L9-1-19 were the only ones that had a higher growth rate in the medium with glucose, that is, the control medium, than in the medium with coffee husk, in addition to expressing the enzymatic activities hydrolase xylanase and cellulase, which suggests a metabolism oriented to the degradation of less complex compounds, thus having more affinity for simpler carbon sources such as glucose.

These are agroindustrial residues that are produced in large quantities in the state of Chiapas, are low cost, and are of biotechnological interest. Coffee husks represent 5 to 12% of the coffee fruit [39], with Chiapas being the main coffee producer with 41% at the national level [40] of about 860 thousand tons of coffee [41]. Chiapas is the main producer of African palm in the country, with 57.3% [42].

Oxidase enzymes such as MnP and LiP are found in Ascomycetes and Basidiomycetes fungi; however, not all of them produce all the enzymes at the same time, or the same quantity of them, and diverse fungi produce different combinations of enzymes, even within the same genus, as occurred among the strains *Trichoderma citrinoviride* TecNM-ITTG L33-2-19, *Trichoderma harzianum* TecNM-ITTG L23-19, *Trichoderma longibrachiatum* TecNM-ITTG L30-2-19 and *Trichoderma reesei* TecNM-ITTG L4-2-19, where one of them did not present MnP activity (Figure 4); or between *Trametes cingulata* TecNM-ITTG L13-19 and *Trametes sanguinea* TecNM-ITTG L14-19, also of the same genus, which had very different LiP production (Figure 5) with 741.770 and 7115.226 U/L, respectively, in addition to having growth rates between 2 and 7 mm/d, indicating that the strain growth rate is not directly related to enzyme production and therefore does not determine its metabolism and degradation mechanisms.

The *Epicoccum sorghinum* TecNM-ITTG L15B-19 strain showed the production of hydrolase enzymes: cellulase using carboxymethyl cellulose as a substrate and xylanase using birch xylan as a substrate to induce the production of enzymes aimed at the degradation of hardwoods. On the other hand, the strains *Trichoderma citrinoviride* TecNM-ITTG L33-2-19, *Trichoderma harzianum* TecNM-ITTG L23-19 and *Trichoderma longibrachiatum* TecNM-ITTG L30-2-19 did not present hydrolase activities: cellulase and xylanase while they had oxidase activities: manganese peroxidase and lignin peroxidase.

The different combinations of ligninolytic enzyme development indicate different abilities of fungi to degrade media containing lignin or other aromatic compounds, which may be related to their biodegradation strategies [43].

## 5. Conclusions

The evaluated fungi that were collected in the state of Chiapas produced hydrolase and oxidase enzymes. The presence of hydrolase (cellulase and xylanase) and oxidase (MnP and LiP) activities in fungi allowed us to classify them into brown-rot fungi, such as the case of the *Aplosporella hesperidica* TecNM-ITTG L30-1-19 strain, and white-rot fungi such as the *Trametes sanguinea* strain TecNM-ITTG L14-19 and the *Phlebiopsis flavidoalba* strain TecNM-ITTG L20-19, representing thirty-nine percent of the isolated strains evaluated.

The fungi growth in PH, CH and PS allowed us to identify a complex metabolism that involves hydrolase and oxidase enzymes.

The species of fungi that were collected in the state of Chiapas are diverse and belong to the ascomycotic and basidiomycotic phylum, which have equivalent enzymatic activities in addition to not being related to the species.

The fact that eighty-nine percent of the species identified represent new records for the state of Chiapas, according to CONABIO data, indicates the importance and relevance of carrying out complete mycological studies for the State of Chiapas, since thanks to the region in which it is located, it has multiple ecosystems that show evidence of the exorbitant prevailing biodiversity.

The present work opens new perspectives for future research regarding the study of lignocellulolytic enzyme-producing fungi and their biotechnological applications.

## Figures and Tables

**Figure 1 jof-07-00450-f001:**
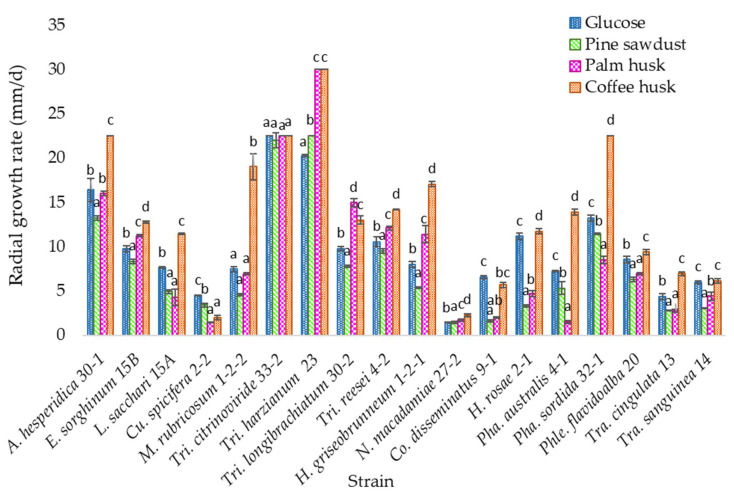
Growth rate in media with lignocellulosic material of the 18 isolates; a, b, c, d: homogeneous groups according to statistical analysis.

**Figure 2 jof-07-00450-f002:**
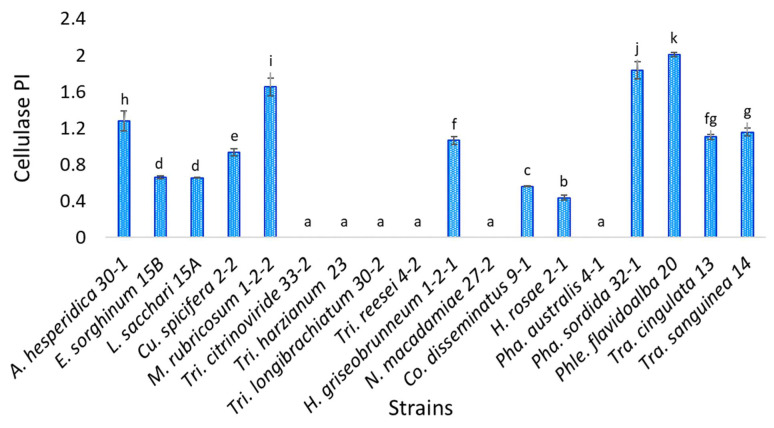
Semi-quantitative cellulase activity of the 18 isolates evaluated; a, b, c, d, e, f, g, h, i, j, k: homogeneous groups according to statistical analysis.

**Figure 3 jof-07-00450-f003:**
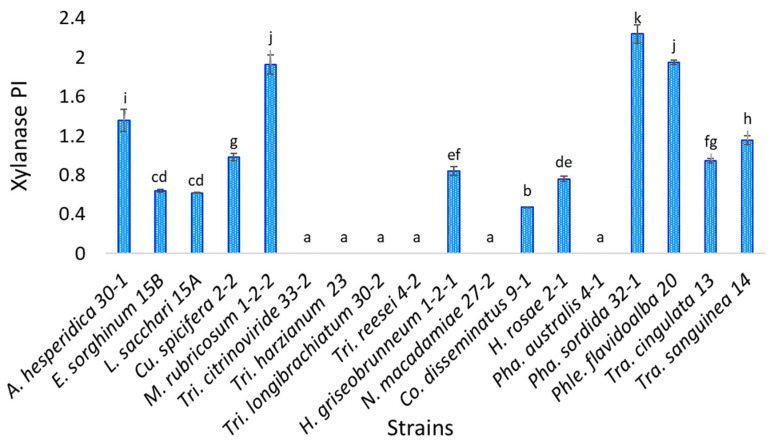
Semi-quantitative xylanase activity of the 18 isolates evaluated; a, b, c, d, e, f, g, h, i, j, k: homogeneous groups according to statistical analysis.

**Figure 4 jof-07-00450-f004:**
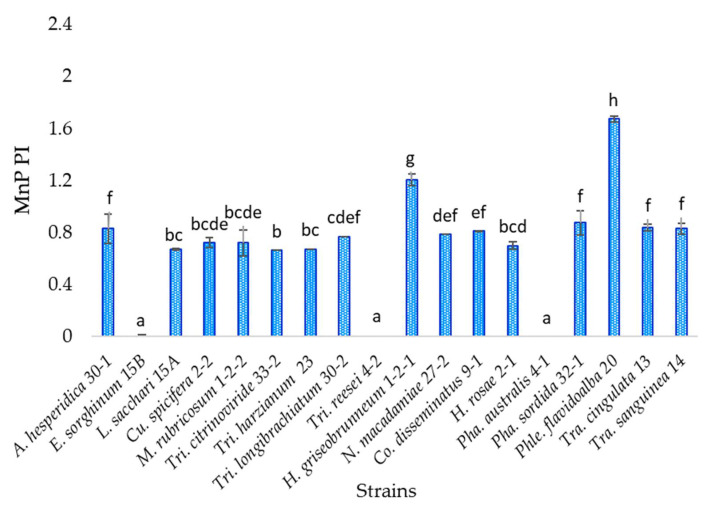
Semi-quantitative manganese peroxidase activity of the 18 isolates evaluated; a, b, c, d, e, f, g, h: homogeneous groups according to statistical analysis.

**Figure 5 jof-07-00450-f005:**
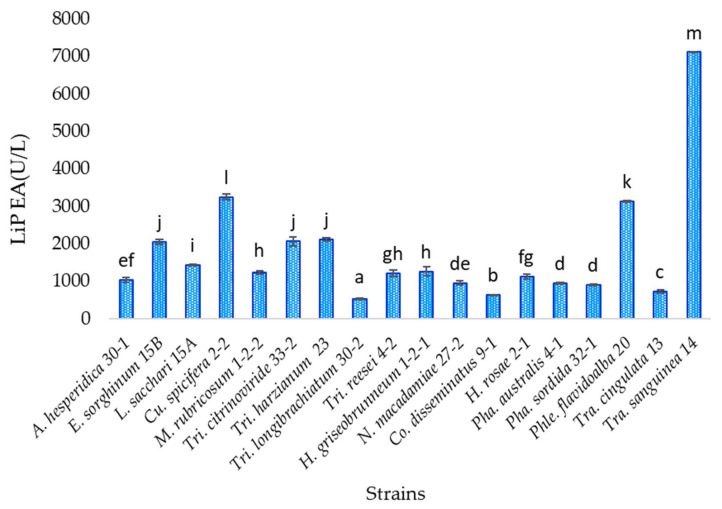
Quantitative manganese peroxidase activity of the 18 isolates evaluated; a, b, c, d, e, f, g, h, i, j, k, l, m: homogeneous groups according to statistical analysis.

**Table 1 jof-07-00450-t001:** Taxonomic category of the identified specimens.

Phylum	Class	Order	Family	Name	Collection Site	Record	Rot Type
Ascomycota	Dothideomycetes	Botryosphaeriales	Aplosporellaceae	*Aplosporella hesperidica* TecNM-ITTG L30-1-19	LVEP ^1^	NR ^4^	BR ^5^
		Pleosporales	Didymellaceae	*Epicoccum sorghinum* TecNM-ITTG L15B-19	SJEP ^2^	NR ^4^	ND ^6^
			Leptosphaeriaceae	*Leptosphaeria spegazzinii* TecNM-ITTG L15A-19	SJEP ^2^	NR ^4^	ND ^6^
			Pleosporaceae	*Curvularia spicifera* TecNM-ITTG L2-2-19	SJEP ^2^	NR ^4^	ND ^6^
		Valsariales	Valsariaceae	*Myrmaecium rubricosum* TecNM-ITTG L1-2-2-19	SJEP ^2^	NR ^4^	ND ^6^
	Sordariomycetes	Hypocreales	Hypocreaceae	*Trichoderma citrinoviride* TecNM-ITTG L33-2-19	LVEP ^1^	NR ^4^	ND ^6^
				*Trichoderma harzianum* TecNM-ITTG L23-19	LVEP ^1^	NR ^4^	ND ^6^
				*Trichoderma longibrachiatum* TecNM-ITTG L30-2-19	LVEP ^1^	NR ^4^	ND ^6^
				*Trichoderma reesei* TecNM-ITTG L4-2-19	SJEP ^2^	NR ^4^	ND ^6^
		Xylariales	Hypoxylaceae	*Hypoxylon griseobrunneum* TecNM-ITTG L1-2-1-19	SJEP ^2^	NR ^4^	ND ^6^
			Sporocadaceae	*Neopestalotiopsis macadamiae* TecNM-ITTG L27-2-19	LVEP ^1^	NR ^4^	WR ^7^
Basidiomycota	Agaricomycetes	Agaricales	Psathyrellaceae	*Coprinellus disseminatus* TecNM-ITTG L9-1-19	SJEP ^2^	-	WR ^7^
		Polyporales	Phanerochaetaceae	*Hyphodermella rosae* TecNM-ITTG L2-1-19	SJEP ^2^	NR ^4^	ND ^6^
				*Phanerochaete australis* TecNM-ITTG L4-1-19	SJEP ^2^	NR ^4^	WR ^7^
				*Phanerochaete sordida* TecNM-ITTG L32-1-19	LVEP ^1^	NR ^4^	WR ^7^
				*Phlebiopsis flavidoalba* TecNM-ITTG L20-19	LVEP ^1^	NR ^4^	WR ^7^
			Polyporaceae	*Trametes cingulata* TecNM-ITTG L13-19	TecNM-ITTG ^3^	NR ^4^	WR ^7^
				*Trametes sanguinea* TecNM-ITTG L14-19	TecNM-ITTG ^3^	-	WR ^7^

^1^ LVEP: “Laguna Verde” Ecotourism Park; ^2^ SJEP: “San José” Educational Park; ^3^ TecNM-ITTG: National Technological of Mexico-Technological Institute of Tuxtla Gutiérrez; ^4^ NR: New Record for the state of Chiapas according to CONABIO data; ^5^ BR: brown rot; ^6^ ND: not determined; ^7^ WR: white rot.

## Data Availability

Not applicable.

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
