# Peer review of "Lignocellulolytic Enzyme Production from Wood Rot Fungi Collected in Chiapas, Mexico, and Their Growth on Lignocellulosic Material"

_jof, 2021, doi:10.3390/jof7060450_

Round 1
Reviewer 1 Report
The authors presented their original research article dealing with the isolation of fungi, their identification and comparing lignocellulolytic enzyme activity. The manuscript fits the scope of the journal, and is interesting but needs some revision. Below I provide my comments and suggestions.
Major suggestions:
- In my humble opinion the introduction part should be improved. Some more new papers should be cited (such as: https://doi.org/10.3390/pr9010038 or https://doi.org/10.3390/pr9020397). The aim of the work should be rephrased and the novelty of the work should be also provided in the introduction.
- Statistical analysis subsection was omitted in the Materials and Methods part.
- In my opinion, changing to coloured bars in Figure 1 will make it easier to read. What about statistical analysis? There is information about statistical differences in the figure caption, but they are not marked on the figure. The same in the Figure 2.
- Section 3.4. Enzyme activities should be rephrased. Every subsection started with information about methodology.
- The discussion part is full of information that may be transferred to the introduction part and were not strictly connected with the results presented in your study. Moreover, the discussion part should be improved and rephrased. Maybe it would be better to combine results and discussion parts, and then more in-depth discussion should be provided.
- The conclusion part should provide some future perspectives about isolated strains not only a summary of your results.
Minor suggestions:
- Some numerical values should be added to the abstract.
-
Figure 2 is illegible in my opinion. Maybe the charts should be enlarged and the points should be changed into bars.
Reviewer 2 Report
The paper is rather long considering that only some fungi from a small geographic region in Mexico were isolated and investigated regarding some wood-degrading enzymes. Already the title of the paper must be modified because also the carbohydrates degrading cellulase and xylanase were measured. The introduction stressed too much on the possible detoxification of poisonous compounds by ligninolytic enzymes whithout that these aspects were ever treated in the following text. Table 1 should also indicate the tree name from which each fungus derived and, if possible, its rot type (WR for white rot, BR for brown rot). Several fungi in Tabe 1 do neither belong to WR or BR, but are moulds may be with some soft-rot activity (e.g. Trichoderma spp.). I also suppose that some isolates are typical litter inhabitants. Figure 2 is difficult to read. The discussion is rather long. I made several remarks/corrections as commentaries in the attached pdf-file.
Olaf Schmidt

Round 2
Reviewer 1 Report
The manuscript has been carefully improved.
One minor change should be corrected in the "2.9 Statistical analysis" section: "p < 0.05" instead of "P> 0.05".
Reviewer 2 Report
My argumentation remains that it was strange to check fungi isolated from softwoods (Pinus spp.) only regarding the hemicellulose xylan degrading xylanases because the main hemicellulose of softwoods is mannan which is degraded by mannanase and ß-mannosidase (general knowledge!!; e.g. Schmidt 2006).
You must declare in the manuscript (Introduction and Discussion) why you used birch xylan for the hemicellulase test and not a softwood mannan!!!
I made some further remarks in the attached pdf.
Olaf Schmidt
